# Bing–Neel Syndrome: Update on Diagnosis and Treatment

Evangeline Y. Wong [1] [iD], Shirley D'Sa [2], Monique C. Minnema [3] [iD], Jorge J. Castillo [4] [iD] and Dipti Talaulikar [1,5,*]

1 School of Medicine and Psychology, College of Health and Medicine, Australian National University, Canberra, ACT 2601, Australia
2 Cancer Division, University College London Hospitals NHS Foundation Trust, London NW1 2BU, UK
3 Department of Hematology, UMC Utrecht Cancer Center, P.O. Box 85500, 2508 GA Utrecht, The Netherlands
4 Bing Centre for Waldenström Macroglobulinemia, Dana-Farber Cancer Institute, 450 Brookline Ave., Boston, MA 02215, USA
5 Department of Haematology, ACT Pathology, Canberra Hospital, Canberra, ACT 2605, Australia
* Correspondence: dipti.talaulikar@anu.edu.au

**Abstract:** Bing–Neel syndrome (BNS) is a rare neurological complication of Waldenström macroglobulinaemia. We highlight key issues in clinical presentation, diagnosis, and treatment while focusing on new and emerging therapies available for patients diagnosed with BNS. It is anticipated that further development of Bruton Tyrosine Kinase (BTK) inhibitors and less toxic chemoimmunotherapies will improve treatment delivery and response.

**Keywords:** Bing–Neel syndrome; Waldenström macroglobulinaemia; ibrutinib; BTK inhibitors; lymphoplasmacytic lymphoma; central nervous system involvement; orbital involvement

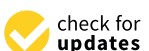



## 1. Introduction

Bing–Neel syndrome (BNS) is a rare complication of Waldenström Macroglobulinaemia (WM), which results from the infiltration of lymphoplasmacytic cells (LPCs) and plasma cells into the central nervous system (CNS). The LPCs and secreted paraprotein can directly damage CNS tissues through direct deposition or through alteration of physicochemical and/or immunological properties.

BNS was first observed in 1936 by Jens Bing and Axel Valdemar von Neel when on autopsies, they identified two cases of LPCs within the CNS of hyperglobulinaemic patients who exhibited CNS signs and symptoms [1]. Interestingly, Bing and Neel's description predated Jan Waldenström's observations of the primary disease, which occurred eight years later in 1944. As further case reports demonstrated the rare complication of WM cells infiltrating the CNS, this condition became known as the Bing–Neel syndrome (BNS).

## 2. Epidemiology

The incidence of BNS is hard to quantify. It is likely an under-recognised condition. Based on the existing literature which consists primarily of case reports and retrospective case series, BNS is estimated to occur in 0.3–1.0% of WM cases. Given that WM is a rare lymphoproliferative disorder with an incidence of approximately three cases per 1,000,000 per year, the limited epidemiological data on BNS are not surprising [2].

While the median age of diagnosis for WM is 63–75 years, the median age at BNS diagnosis is approximately 61 years (range 36–82) [3]. Apart from a diagnosis of WM, there have been no other risk factors identified for BNS [4]. BNS is more commonly reported in males, with two retrospective studies reporting 56% and 80% males in a total cohort of 34 and 44 BNS patients, respectively [5,6].

BNS can occur as the presenting feature of WM or occur later during the course of the disease. The median time to BNS among patients with a known diagnosis of WM is 8.9 years [6]. However, two retrospective studies found that 15–36% of patients were diagnosed with BNS as their first manifestation of the disease without prior history

of WM [5–7]. Reports suggest these patients may have a better prognosis than those presenting later [6]. It is important to recognise that BNS may develop in the setting of asymptomatic/untreated WM, and in those who have achieved a response following treatment for WM (even those in complete remission) [6]. If CNS involvement occurs in the setting of high-grade transformation of WM, careful diagnostic tests should be undertaken to establish the histological nature of the CNS involvement, as this may influence treatment approaches.

### 3. Pathophysiology

Understanding the pathophysiology of BNS remains an active area of research. With better recognition and understanding of the condition over recent decades, it is believed that infiltration of WM monoclonal LPCs and plasma cells disrupts CNS parenchyma [8–10]. This results in nerve demyelination, axonal degeneration, CNS toxicity, and reactive gliosis, all of which ultimately impair CNS function.

### 4. Clinical Presentation

The diagnosis of BNS remains challenging in those without a prior diagnosis of WM, as the signs and symptoms (Table 1) are heterogeneous and may be seen in many aetiologies. In patients with a history of WM, BNS should be actively screened for when unexplained neurological signs and symptoms develop. A major challenge in such situations may be recognising neurological manifestations attributable to BNS and differentiating them from other complications associated with WM, such as hyperviscosity syndrome, IgM-related neuropathy or circulatory disturbances [11].

**Table 1.** Clinical features of Bing–Neel syndrome.

| Clinical Features |
| --- |
| • Ataxia and balance disorders (48%) |
| • Cranial nerve involvement (36%) |
| • Cognitive impairment (27%) |
| • Paresis and motor or sensory symptoms (25%) |
| • Headache (18%) |
| • Cauda equina syndrome (14%) |
| • Visual impairments/disturbances |
| • Hearing deficits |
| • Psychiatric symptoms |

The neurological manifestations in BNS usually develop progressively over weeks to months. Based on the two largest retrospective BNS case series available, a summary of 78 reported patients showed that balance disorder or ataxia was found in 12–48%, altered mental status in 27–35%, motor deficits, especially of the limbs in 14–35%, and cranial nerve involvement in 29–36% (refer to Table 1) [5,6]. In a single-centre study, Simon et al. observed mainly oculomotor and facial nerves implicated in cranial nerve involvement [5]. It should be highlighted that presentation of BNS can occur in patients with peripheral neuropathy.

As with other CNS structures, WM neoplastic cells or IgM paraprotein products can infiltrate any part of the eye. For example, infiltration into the anterior segment can mimic iritis or vitritis. The cells or secreted products may also infiltrate the choroid, retina, extraocular muscles, or optic nerve. Decreased visual acuity, visual disturbances or orbitopathy can be a feature of BNS [12]. Infiltration into the optic nerve sheath, which is surrounded by cerebrospinal fluid (CSF), can potentially lead to neoplastic meningitis or increased intracranial pressure leading to papilloedema [13,14].

## 5. Diagnostic Findings

### 5.1. Radiological Features

The most common sites for WM and macroglobulin infiltration are the meninges and CSF (leptomeningeal involvement), followed by cerebral parenchyma and spinal cord lesions. In most patients presenting with neurological complaints, neuroimaging forms part of the core diagnostic work-up of BNS.

While computerised tomography (CT) scans can identify intracranial masses and leptomeningeal involvement, magnetic resonance imaging (MRI) is the gold-standard imaging modality for the diagnosis of BNS. It is recommended that MRI brain and spine fluid-attenuated inversion recovery (FLAIR) and T1-weighted sequences before and after gadolinium be obtained (see Table 2). Importantly, radiological imaging should be conducted before lumbar puncture as the procedure could create artefacts leading to false positives on MRI scans. Similarly, if possible, imaging should be performed before steroid administration.

**Table 2.** Summary of assessment of patients with suspected Bing–Neel Syndrome.

| Assessment |
| --- |
| Radiology |
| • MRI is the gold-standard imaging modality for suspected BNS. |
| • MRI brain and spine should be performed before lumbar puncture to avoid artefacts causing false positive findings. |
| Cytology/histology and ancillary investigations |
| • The presence of WM cells (LPCs) in CNS tissue or CSF is required for the diagnosis of BNS. |
| • WM LPCs are small, atypical B lymphocytes, often with plasmacytic differentiation, that do not meet the criteria of other small B-cell disorders. Plasma cells may be observed. |
| • The characteristic immunophenotypic features of WM cells are surface light chain restriction, pan-B cell surface markers (CD19, CD20, CD22) with IgM$^+$ and variable plasma cell markers CD38 and CD138. They can have variable expression of CD27, CD52, CD5, and CD3. |
| • Molecular analysis for MYD88$^{L265P}$ mutation and Ig heavy chain rearrangement aid in the diagnosis of BNS. Identifying these in the CNS/CSF and bone marrow provides evidence of clonality. |
| Additional points |
| • Consider and exclude hyperviscosity syndrome as a cause of patient's neurological complaints. |
| • For suspected high-grade transformation, CNS tissue biopsy may be essential. |
| • For patients newly diagnosed with WM, basic investigations for WM disease will include bone marrow aspirate and trephine biopsy, immunophenotypic assessment, molecular analysis for MYD88$^{L265P}$, and paraprotein assessment with serum protein electrophoresis and immunofixation +/- biopsy of extramedullary site. |

MRI—magnetic resonance imaging; BNS—Bing–Neel Syndrome; WM—Waldenström Macroglobulinaemia.

The most common radiological finding observed in a cohort of 24 patients diagnosed with BNS was a subarachnoid enhancement in T1 images (71%) which suggests leptomeningeal infiltration [15]. The study found that cauda equina enhancement represents the most common site of spinal nerve involvement. Enhancement and thickening indicative of dural involvement are best viewed in contrast-enhanced T1 or FLAIR images. Diffusion-weighted MRI can assist in identifying vasogenic oedema, which may add weight to the diagnosis of BNS over hyperviscosity syndrome [16]. Localisation of lesions in the brain parenchyma is less common and if found, lesions are usually located in periventricular/subependymal regions. Radiologically, these are best visualised on T2 and FLAIR images as hyperintensities, which likely correspond to the effects of blood-brain barrier (BBB) disruption in the early stage or demyelination in later stages [15].

MRI can also identify CNS tissue amenable to biopsy, which remains the gold standard in diagnosing BNS.

*5.2. Laboratory Features*

5.2.1. Testing for WM

Routine laboratory findings usually reveal features of WM [17], though it is possible for patients to have positive findings on CNS biopsy and CSF without systemic involvement by WM. At a minimum, blood work-up should include full blood count, basic biochemistry including renal function +/- LDH, +/- serum albumin, serum immunoglobulin levels (IgM, IgG, IgA), β2 microglobulin, serum protein electrophoresis including immunofixation, serum viscosity if available, +/- cryoglobulins. If a diagnosis of WM is not already established, bone marrow biopsy, along with ancillary tests, is required to confirm bone marrow infiltration with clonal LPCs and plasma cells.

5.2.2. Testing for BNS

Current guidelines for definitive BNS diagnosis recommend histologic biopsy of CNS tissue or identification of LPCs with B cell monoclonality in CSF and no evidence of transformation [18]. In practice, obtaining a tissue biopsy is not always feasible, therefore, CSF examination is considered key to diagnosis [19]. Diagnostic accuracy of CSF can be improved through correct lumbar puncture technique and optimal and adequate collection of CSF. Adequate volume is needed for cytology, flow cytometry, molecular analysis, and culture if infection is a differential diagnosis. In situations of a bloody tap, clinicians should collect multiple specimens and use the last tube for flow cytometry and molecular analysis. This helps to minimise potential false positives in interpreting circulating LPCs (WM cells) from a blood tap as LPCs in the CSF. Interestingly, there have been reports of normal CSF examinations in which cytological confirmation of BNS was obtained later [5,20]. Therefore, a negative CSF result does not rule out BNS as this could reflect a low disease burden leading to cells not being collected during CSF sampling.

It is essential to demonstrate WM cells. The diagnosis cannot currently be made by measuring M protein in the CSF as a disrupted BBB may allow IgM seepage from the bloodstream. What may be helpful for the diagnosis of BNS in cases where WM cells cannot be demonstrated is the IgM index which is calculated by [CSF IgM (mg/L)/serum IgM (g/L)]/[CSF albumin (mg/L)/serum albumin (g/L)]. An elevated IgM index (normal reference range <0.060) likely indicates that the concentration of IgM monoclonal protein in CSF supersedes what would have been expected from passive leakage across the BBB [21]. This alludes to the presence of WM cells in the patient's CSF.

If the CSF does not enable definitive diagnosis via flow cytometry or molecular studies (see further sections and Figure 1), a brain biopsy may be needed. However, this is dependent on the location of the target lesion and the procedural risk. In the setting of suspected high-grade transformation, a brain biopsy may be essential.

Morphology

Cytology of CSF or CNS tissue biopsy are characterised by small, atypical B lymphocytes, i.e., LPCs with plasmacytic differentiation, including cytoplasmic basophilia and eccentric nuclei, that do not meet the criteria of other small B-cell disorders [22]. Plasma cells can also be observed.

Immunophenotyping

The clonal nature of the malignant cells in the CNS is confirmed with flow cytometry. The immunophenotype characteristic of BNS includes positive B cell surface markers (CD19, CD20, CD79a, CD79b), positive plasma cell markers (CD138 and CD38), and variable CD27, CD52, CD5, CD3 expression (see Figure 1) [18,23]. Kappa light chain restriction is usually observed though lambda light chain restriction has been occasionally reported [5,6,19,20].

Cytomorphology
- Visualising WM cells in CNS (histology of CNS tissue or cytology of CSF) is the gold standard for diagnosis
- Small atypical malignant B cells mixed with LPCs and plasma cells

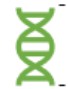

Immunophenotyping
- Pan B cell markers (CD19, CD20, CD79a, CD79b), plasma cell markers (CD138, CD38) and IgM
- Variable CD25, CD27, CD52, CD5, CD3
- Kappa light chain restriction > lambda light chain

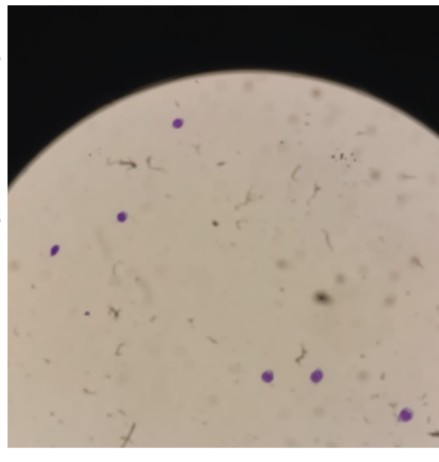

CSF cytospin showing LPCs

Molecular genetics
- Detection of somatic mutation MYD88$^{L265P}$ mutation
- Establish clonality using Ig heavy chain rearrangement

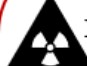

Radiological features
- MRI is gold-standard
- High T2 signal in brain parenchyma, subarachnoid enhancement, leptomeningeal thickening/enhancement, cauda equina enhancement, non-specific vasogenic oedema
- Optic pathway compression, increased T2 signal in orbital structures

**Figure 1.** Summary of diagnostic features seen in patients with Bing–Neel Syndrome.

Molecular Features

Conventional karyotyping has a limited role in BNS given the low rate of cell proliferation. Recent advances in molecular diagnostics provide more sophisticated ways of identifying LPC infiltration into the CNS and supporting the diagnosis of BNS. Specifically, analysing for MYD88[L265P] mutation and Ig heavy chain rearrangement via polymerase chain reaction (PCR) assays has high sensitivity when they are identified in both bone marrow and CSF [18]. This is particularly useful when patients have already received steroid therapy to suppress symptoms causing the destruction of WM cells which are then not morphologically apparent.

In 2012, whole genome sequencing identified the presence of a single point mutation leading to a leucine-to-proline amino change at position 265 of the myeloid differentiation factor 88 gene (MYD88) located at chromosome 3p22.2 [24]. This provides diagnostic utility for both WM and BNS. The somatic mutation MYD88[L265P] can be detected in the CSF of >90% of BNS patients (see Figure 1). In fact, a study on 34 patients diagnosed with BNS found that MYD88[L265P] was present in 100% of their cohort [6]. However, MYD88[L265P] is not specific for BNS as it is also present in some cases of Primary CNS Lymphoma (PCNSL), as well as Chronic lymphocytic leukaemia (CLL) and Marginal zone lymphoma (MZL) that may involve immune-privileged sites [25]. Nevertheless, detecting MYD88[L265P] in the CSF is useful in supporting the diagnosis of BNS in the appropriate context. Furthermore, quantifying MYD88[L265P] with qPCR in the CSF can potentially provide a useful molecular tool in monitoring response to treatment.

Molecular analysis of Ig gene rearrangement in the CSF can provide confirmation of clonality [26,27]. Clonal IgH rearrangements that share the same nucleotide sequence in CSF (or CNS tissue) and bone marrow samples strongly support the diagnosis of BNS. A cohort of 34 patients with BNS had a 94% detection rate of IGH rearrangement [5].

## 6. Differential Diagnosis

Although neurological complaints may reflect infiltration of LPCs into the CNS, a significant proportion of neurological complaints in patients with WM may be attributed to hyperviscosity (see Table 3). Patients diagnosed with WM-associated hyperviscosity have neurological features, including blurred vision, visual loss, ataxia, headache, or dizziness, and altered cognition [28–30]. Obtaining serum IgM levels and serum viscosity, if available, are important in establishing the diagnosis of hyperviscosity. Serum viscosity of ≥4 cp should prompt consideration of hyperviscosity syndrome rather than BNS.

Approximately 20% of patients present with neuropathy at the time of their WM diagnosis [31,32]. Many of these are explained by anti-myelin-associated glycoprotein [33] activity rather than CNS infiltration [34]. These patients exhibit a slowly progressive sensorimotor peripheral neuropathy (over years) that commonly involves distal limbs in a symmetrical manner. In contrast, the neurological manifestations of BNS patients tend to develop progressively over weeks to months and tend to be dominated by central rather than peripheral nerve involvement.

Other differentials for BNS include other brain tumours or alternate lymphoma/leukaemia involving the CNS. Histopathology analysis of tissues is required to differentiate BNS from other malignancies.

**Table 3.** Differential diagnoses for BNS.

| Differential Diagnosis | Comments |
|---|---|
| Hyperviscosity syndrome | • Similar neurological S/Sx to BNS, including visual problems, headache/dizziness.<br>• Mucosal bleeding at presentation indicates hyperviscosity, not BNS.<br>• Serum viscosity >4 centipoises favours hyperviscosity syndrome.<br>• Circulatory disturbances from hyperviscosity can be confirmed with fundoscopy examination showing distended/tortuous retinal veins, papilloedema, haemorrhages [35]. |
| Anti-MAG antibodies | • Neuropathy from Anti-MAG antibodies is usually distal and symmetrical, with sensory and motor deficits that slowly progress (years) vs relative faster period (months–years) in BNS [36].<br>• Most anti-MAG antibodies are IgMκ [37]. |
| Other cancers involving the CNS | • Various other brain tumours and leukaemia, and lymphomas can involve the CNS.<br>• Cytological analysis with immunohistochemistry and multiparametric flow cytometry are required for accurate diagnosis [38]. |

S/Sx—signs/symptoms; BNS—Bing–Neel Syndrome; MAG—myelin-associated glycoprotein; CNS—central nervous system.

## 7. Therapeutic Strategy

### 7.1. General Approach

Treatment is indicated in symptomatic patients with a confirmed diagnosis of BNS [18]. There is a paucity of high-level evidence on treating patients with BNS. The current consensus on treatment is based on case reports and retrospective case series, and most recently, on the recommendations of the 8th International Workshop for Waldenström Macroglobulinaemia Task Force on Bing–Neel syndrome [18,19].

The principle and success of treatment depend on its ability to penetrate the BBB to target malignant cells within the CNS. Therefore, standard treatments used in WM may have limitations in the treatment of BNS. The goal of treatment is to relieve symptoms and induce prolonged progression-free survival (PFS).

Since WM is a non-curable disease, striving for clearance of LPCs in the CNS of BNS patients is possibly futile. Indeed, some patients may continue to have detectable disease on CSF despite having a reversal of their symptoms [6]. Furthermore, radiological abnormalities that persist post-treatment could reflect either radiological lag relative to clinical symptoms or gliosis or demyelination of tissue rather than the active disease [18]. Ongoing radiological abnormalities post-treatment do not necessarily imply active BNS disease.

Treatment of BNS is individualised for each patient, based on key considerations such as site and degree of infiltration, functional impairment, and patient fitness.

Broadly speaking, treatment options for BNS encompass intravenous (IV) chemotherapy, intrathecal (IT) chemotherapy, Bruton kinase inhibitor (BTKi), purine analogues (Fludarabine, Cladribine), Bendamustine, and radiotherapy (see Figure 2).

### 7.2. Chemotherapy

#### 7.2.1. High-Dose (HD) Chemotherapy: HD-Methotrexate, HD-Cytarabine

Traditionally, treatments for BNS have been adopted from treatment regimens used in PCNSL, such as HD-Methotrexate and HD-Cytarabine. Although reports have outlined their effectiveness in treating BNS patients, they cause side effects that may be disproportionate to the benefit obtained. However, if the BNS is responsible for reduced mobility and hence poor performance status, more intensive approaches may be justified to accelerate neurological recovery and functionality and contribute to net long-term benefits. A case-by-case assessment is required, taking account of disease bulk in the CNS, age and fitness of the patient, the burden of systemic disease (which may need controlling and careful discussion with the patient of the possible risks and benefits of more intensive therapies) (see Table 4).

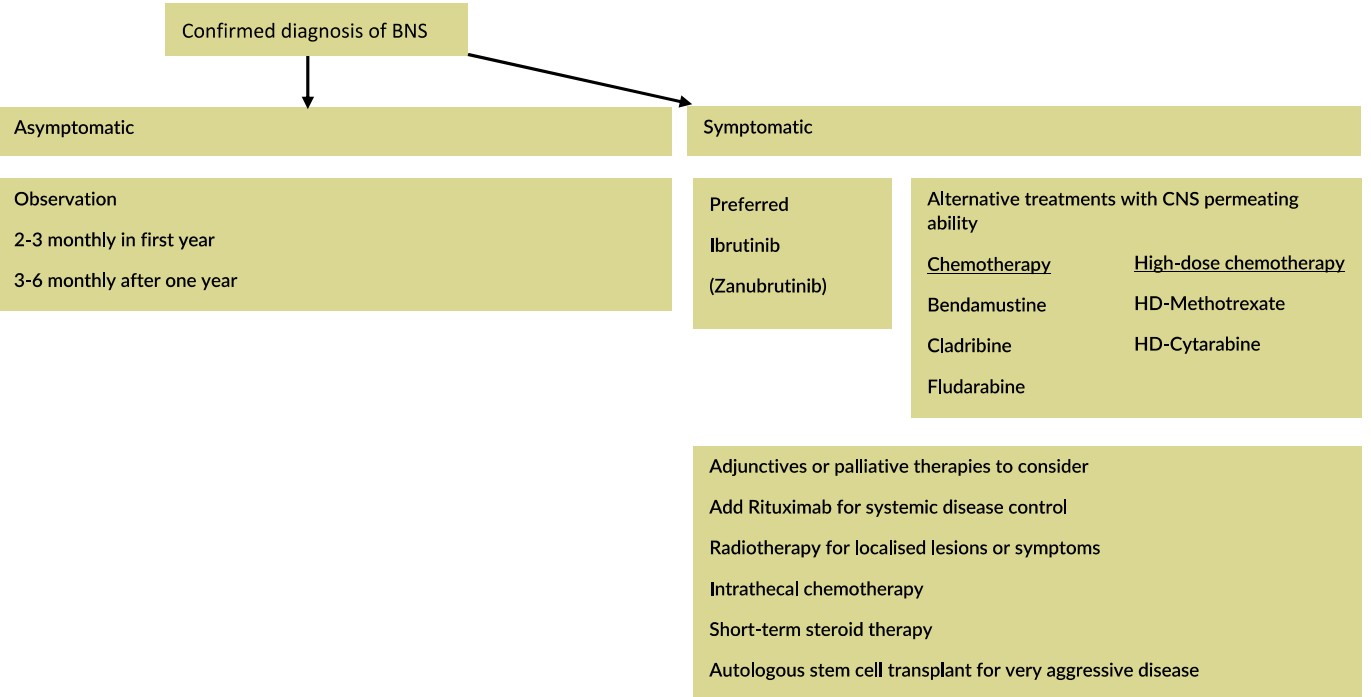

**Figure 2.** Therapeutic strategy for confirmed diagnosis of BNS. BNS—Bing–Neel syndrome; HD—high-dose. Adapted from Ref. [18].

**Table 4.** Pretreatment considerations in the management of Bing–Neel syndrome.

| Pretreatment Considerations |
| --- |
| • Presence of neurological symptoms |
| • Patient age, fitness, medical co-morbidities |
| • Prior lymphoma therapy, including types of therapy, quality, and duration of initial response |
| • Site and bulk of CNS disease |

CNS—central nervous system.

### 7.2.2. Bruton Tyrosine Kinase Inhibitors (BTKi)

When BTKi were recognised to cross the BBB, they became potential treatment agents for patients with BNS [39–41]. Indeed, an international multicentre retrospective study on 28 patients demonstrated that single-agent ibrutinib at a dose of 560 mg (46%) or 420 mg (54%) once daily was effective in patients with BNS with 85% of the patient cohort achieving symptomatic improvement, and 60% showing radiological improvement within three months of therapy [20]. Two-year event-free survival rate with ibrutinib was 80% (95% CI 63–95%). At a median follow-up of one year, the dose of Ibrutinib did not have an impact on treatment response. The study recommends commencing patients on 420 mg and during interim assessment, if there is no response, increasing it to 560 mg.

Zanubrutini—a second-generation BTKi– has been shown to be highly potent and specific in irreversibly binding to BTK [42]. One case report of a 75-year-old patient describes the positive effect of 15 months of 160 mg twice daily of Zanubrutinib used for treatment of WM on improving neurological symptoms and functionality after cycles of high-dose of IV Methotrexate failed to achieve significant MRI response [43]. In the treatment of WM, head-to-head BTKi trial comparing Zanubrutinib with Ibrutinib demonstrated higher rates of complete response and very good partial response in patients in the Zanubrutinib arm, with good long-term safety and tolerability at a median follow-up of 43 months [44]. With a greater understanding of the efficacy of Zanubrutinib, Zanubrutinib may soon have a role in treating BNS patients.

### 7.2.3. Chemotherapy: Fludarabine, Cladribine, and Bendamustine

Fludarabine and Cladribine are structurally related purine nucleoside analogs. Fludarabine has shown utility in treating BNS with the advantage of oral administration. In a study of four consecutive patients presenting with a total of five episodes of BNS, two out of four patients achieved partial response to first-line therapy made up of six cycles of Fludarabine-Rituximab based chemoimmunotherapy +/- IT Methotrexate [45]. One patient had complete response from Fludarabine-Rituximab plus regular 15 mg IT Methotrexate, and the other patient required Fludarabine as second-line therapy to achieve complete remission. Cladribine became a potential treatment for BNS when 26 patients with newly diagnosed WM showed an 85% response rate to Cladribine [46]. Cladribine's utility in BNS was also demonstrated in case reports of Cladribine monotherapy in one patient and Cladribine/Cyclophosphamide/Prednisone with radiotherapy in another patient [47,48]. Cladribine's ability to penetrate the CNS is shown in pharmacokinetic studies demonstrating CSF drug levels exceeding plasma levels in patients with meningeal disease [49]. How and when to employ purine analogues in the BNS setting remains unclear. They could have a role in those unsuitable for high-dose chemotherapy or at relapse. In the rare occasion where autologous stem cell transplant is being considered for aggressive WM or BNS, purine analogues should be avoided because of the risk of stem cell toxicity.

Although WM cells express CD20, Rituximab monotherapy is not advised as its CNS penetrance is unclear [50]. Rituximab-Bendamustine on a 28-day cycle plus 15 mg IT Methotrexate with each cycle was studied in a medically and functionally poor patient who developed BNS while receiving chemoimmunotherapy for WM. The post-treatment MRI demonstrated a nearly complete disappearance of leptomeningeal enhancement [51]. Side effects of Bendamustine include risk of developing myelodysplastic syndrome (MDS) or acute myeloid leukaemia (AML) and myelosuppression [52].

### 7.2.4. Intrathecal Methotrexate

IT Methotrexate may be administered at a usual dose of 15 mg in BNS, but there are no definitive data for long-term benefits as a single agent [14]. Currently, there is no consensus regarding the usefulness of IT chemotherapy in the BNS setting.

### 7.3. Radiotherapy

BNS is radiosensitive, which makes radiotherapy an effective treatment option for localised CNS involvement [10,53]. It is important to remember that total brain irradiation may result in cognitive impairment, especially in the elderly population [54,55]. Given its neurotoxicity, aside from localised spinal disease, radiotherapy is not recommended as a first-line therapy and should be reserved for patients who fail other treatment options [18].

### 7.4. Autologous Stem Cell Transplant (ASCT)

At present, it is unclear whether autologous stem cell transplant (ASCT) should be an option for symptomatic patients with BNS. Given that few BNS patients have received ASCT, it is not possible to comment on optimal conditioning therapy [5,6,56,57]. Provided the patient is medically and functionally fit, ASCT can be considered for certain patients, such as younger patients with BNS and aggressive systemic disease. ASCT provides a good long-term response in which 79% of a cohort of 14 BNS patients remained relapse-free at three-year follow-up [57].

### 7.5. Management of Orbital BNS

BNS involving the eye and surrounding structures deserve special mention as there are additional complexities in both diagnosis and management. Neuro-ophthalmology input is paramount in the patient's assessment and management. Decreased visual acuity can be due to the involvement of the extraocular muscles, optic nerve, chiasm, optic tract or beyond. For lesions located in surgically inaccessible sites, the use of conjunctival biopsy should be considered a minimally invasive method for assessing eye involvement.

The use of conjunctival biopsies as a surrogate has been described in other diseases [58]. Visualising infiltration of malignant WM cells in tissue biopsy is key to the diagnosis, and if not surgically resectable, strenuous efforts should be made to confirm evidence of disease via complementary means (e.g., with CSF).

To date, only a few cases have been reported in the literature with variable treatment regimens [12,59–62]. Briefly, five out of six reports available utilised some form of IT chemotherapy, more commonly IT methotrexate than IT Cytarabine. IT therapy was combined with systemic chemoimmunotherapy: IT Methotrexate with R-CHOP, IT Methotrexate with Bendumustine with Rituximab, or IT Cytarabine with IV Methotrexate [12,61,62].

One case did not use IT chemotherapy but rather systemic chemoimmunotherapy including Cyclophosphamide, Prednisolone, and Rituximab, which led to improvement in optic nerve function and radiological regression [59]. Radiotherapy may have a palliative or consolidative role in treating BNS involving the eye. A case of palliative orbital radiation was described whereby 10 fractions of radiation to a total dose of 20Gy was used [60].

## 8. Monitoring Response and Interim Assessment

Treatment response criteria for BNS as proposed by the task force is summarised in Table 5 [18]. Clinicians should consider undertaking an interim assessment to evaluate patient's treatment response. If starting with a more intensive regimen, an interim assessment (e.g., after two cycles) could enable consolidation with a less intensive approach, such as a BTKi. Patients on continuous treatment should be reviewed clinically with laboratory and radiology every three to four months and consider spacing our MRIs if clinically stable [18]. Patients showing neurological improvements can potentially transition to a 12-monthly review after the first year of treatment.

**Table 5.** Treatment response criteria for Bing–Neel Syndrome.

| Treatment Response | Comments |
|---|---|
| Complete remission (CR) | Resolution of clinical symptoms with normalisation of CSF, and normalisation of MRI findings (or MRI demonstrating minimal residual abnormalities on T2 or FLAIR). |
| Partial response (PR) | Improvement in clinical symptoms with no complete resolution OR; Complete resolution of clinical symptoms with persisting radiological abnormalities (not including minimal residual abnormalities on T2 or FLAIR) There should be normalisation of CSF. |
| Non-response (NR) | Progression or persistence of clinical symptoms, CSF and radiological findings. |
| Relapse | New signs or symptoms reappearing because of BNS OR; Radiological progression +/- new radiological findings OR; BNS disease detected by cytology, flow cytometry, and/or molecular analysis. |

CSF—cerebrospinal fluid; MRI—magnetic resonance imaging; FLAIR—fluid-attenuated inversion recovery.

As previously mentioned, the goal of treatment is to restore neurological function and improve patient's quality of life. CSF markers depicting down trending towards normalisation provide a useful way to monitor response. Specifically, sequentially monitoring MYD88$^{L265P}$ provides good indication of treatment response and disease progression [63]. As with WM patients, patients who develop BNS should continue to be monitored by the Haematology team.

## 9. Disease Prognosis

Currently, there are no recognised prognostic factors for BNS. Based on published case reports and series, patients who were diagnosed with BNS as the first manifestation of WM appear to do better than patients who develop BNS later in the disease course [6]. Overall, case series reports found a survival rate of 71% at five years and 59% at 10 years

post diagnosis of BNS [5]. Specifically, at three years, 40% of survivors showed either pathological or radiological persistence of the disease [6].

With BTKi being increasingly recognised as an effective treatment for BNS, overall survival and quality of life are likely to improve.

## 10. Conclusions

BNS is a rare but important neurological complication of WM that usually occurs during disease progression, although not necessarily in the context of systemic relapse. A high level of suspicion is prudent when assessing patients with WM who develop neurological signs and symptoms. Many treatment options are effective in the BNS setting. Their use should be tailored to the patient's individual circumstances, including fitness and co-morbidities and guided by patient-specific treatment goals.

**Author Contributions:** E.Y.W. wrote the primary draft under D.T. supervision. S.D., J.J.C. and M.C.M. reviewed the draft and provided input. All authors have read and agreed to the published version of the manuscript.

**Funding:** This research received no external funding.

**Institutional Review Board Statement:** Not applicable.

**Informed Consent Statement:** Not applicable.

**Data Availability Statement:** No new data were created or analysed in this study. Data sharing is not applicable to this article.

**Conflicts of Interest:** The authors declare no conflict of interest.

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
