# Peer review of "Bing–Neel Syndrome: Update on Diagnosis and Treatment"

_hemato, doi:10.3390/hemato3040051_

Round 1
Reviewer 1 Report
(General comments)
Wong et al wrote the comprehensive review of the Bing-Neel syndrome (BNS). The manuscript was well written, and informative. I have 3 minor concerns as bellow.
(Comments)
1.In 5.2. Laboratory features, it is essential to demonstrate WM cells themselves; the diagnosis cannot currently be made by measuring M protein in the CSF as a disrupted BBB may allow IgM seepage from the bloodstream (P4, line 135-137).
I agree with the idea of M-protein in the CSF alone is not sufficient for making precise diagnosis of BNS, however, IgM index calculated by“[CSF IgM (mg/L) / serum IgM (g/L)] /[CSF albumin (mg/L) / serum albumin (g/L)]” is helpful for the diagnosis of BNS. Thus, could you comment the IgM index in 5.2. Laboratory features section.
2.In Table 1, the word “headache” is written in 2 places, and it is confusing.
Cognitive impairment, headache (27%)
Headache (18%)
Please check the Table is correct.
3. Many typos of MYD88L264P (MYD88L265P) are found in 5.2.3. Molecular features, Fig1, and Table2.
Reviewer 2 Report
This manuscript offers a complete review of the literature on BNS.
Some remarks:
- the paper would benefit from a general editing of English language and style
- line 30: occurred (not occurr)
- line 40 : apart from having (not apart from a having)
- line 41 there HAS been
- line 57: plasma cells disrupt (not plasma cells and disrupt)
- line 58 and 107 : parenchyma (not parenchymal)
- table 1: some features are reported twice (headache, paresis and motor symptoms/motor deficits); visual impairment OR disturbances (not VISUAL IMPAIRMENT OF DISTURBANCES)
- in the whole manuscript/figures/tables: MYD88 L265P and not MYD88 L264P
- line 223 will guide (not will help guide)
- line 224: WM not WN
- line 277: once BTK was recognised (not once BTKis were)
